# *Chlamydia buteonis* in birds of prey presented to California wildlife rehabilitation facilities

Brittany A. Seibert[1¤], Michael K. Keel[2�], Terra R. Kelly[3], Roger A. Nilsen[4],
Paula Ciembor[4], Denise Pesti[4], Christopher R. Gregory[4], Branson W. Ritchie[4☉], Michelle
G. Hawkins[5☉] *

1 Department of Avian Science, University of California, Davis, California, United States of America,
2 Department of Pathology, Microbiology, and Immunology, School of Veterinary Medicine, University of
California, Davis, California, United States of America, 3 Karen C. Drayer Wildlife Health Center, School of
Veterinary Medicine, University of California, Davis, California, United States of America, 4 Infectious Disease
Laboratory, University of Georgia, Athens, Georgia, United States of America, 5 Department of Medicine and
Epidemiology, School of Veterinary Medicine, University of California, Davis, California, United States of America

☉ These authors contributed equally to this work.
¤ Current address: Department of Population Health, College of Veterinary Medicine, University of Georgia,
Athens, Georgia, United States of America
* mghawkins@ucdavis.edu

**Data Availability Statement:** All relevant data are within the manuscript.

## Abstract

Chlamydial infections, caused by a group of obligate, intracellular, gram-negative bacteria, have health implications for animals and humans. Due to their highly infectious nature and zoonotic potential, staff at wildlife rehabilitation centers should be educated on the clinical manifestations, prevalence, and risk factors associated with *Chlamydia* spp. infections in raptors. The objectives of this study were to document the prevalence of chlamydial DNA shedding and anti-chlamydial antibodies in raptors admitted to five wildlife rehabilitation centers in California over a one-year period. Chlamydial prevalence was estimated in raptors for each center and potential risk factors associated with infection were evaluated, including location, species, season, and age class. Plasma samples and conjunctiva/choana/cloaca swabs were collected for serology and qPCR from a subset of 263 birds of prey, representing 18 species. Serologic assays identified both anti-*C. buteonis* IgM and anti-chlamydial IgY antibodies. Chlamydial DNA and anti-chlamydial antibodies were detected in 4.18% (11/263) and 3.14% (6/191) of patients, respectively. Chamydial DNA was identified in raptors from the families Accipitridae and Strigidae while anti-*C.buteonis* IgM was identified in birds identified in Accipitridae, Falconidae, Strigidae, and Cathartidae. Two of the chlamydial DNA positive birds (one Swainson's hawk (*Buteo swainsoni*) and one red-tailed hawk (*Buteo jamaicensis*)) were necropsied, and tissues were collected for culture. Sequencing of the cultured elementary bodies revealed a chlamydial DNA sequence with 99.97% average nucleotide identity to the recently described *Chlamydia buteonis*. Spatial clusters of seropositive raptors and raptors positive for chlamydial DNA were detected in northern California. Infections were most prevalent during the winter season. Furthermore, while the proportion of raptors testing positive for chlamydial DNA was similar across age classes, seroprevalence was highest in adults. This study questions the current knowledge on *C. buteonis* host range and highlights the importance of further studies to evaluate the diversity and epidemiology of *Chlamydia* spp. infecting raptor populations.

**Funding:** This study was financially supported through the Karen C. Drayer Wildlife Health Center Fellowship Grant Program, School of Veterinary Medicine, Univ. of California, Davis, Davis, CA and the Infectious Disease Laboratory, College of Veterinary Medicine, University of Georgia, Athens, GA. The Karen C. Drayer Wildlife Health Center funders had no role in study design, data collection and analysis, decision to publish, or preparation of the manuscript. The Infectious Disease Laboratory, University of GA donated the costs of the qPCR testing, EBA testing and chlamydial cultures and were instrumental in data analysis, decision to publish and preparation of the manuscript. Multiple authors are affiliated with the laboratory (Ritchie, Nilsen, Ciembor, Pesti, Gregory) but only Ritchie and Nilsen were involved in manuscript preparation.

**Competing interests:** Dr. Branson Ritchie is the Director of the Infectious Disease Laboratory, College of Veterinary Medicine, Univ. of GA, Athens, GA and contributed in-kind support by providing the EBA serology, qPCR testing and whole genome sequencing for this project.

## Introduction

*Chlamydiae* are obligate intracellular, gram-negative bacteria responsible for a variety of diseases in multiple species, including humans [1]. One example, *Chlamydia psittaci*, is a cosmopolitan chlamydial species and important zoonotic pathogen. This organism has a global distribution and has been reported in > 469 bird species encompassing 30 orders [2,3]. Currently, there are 14 recognized and 3 candidate species in the genus *Chlamydia* [4–9], including the newly identified *C. buteonis* [10].

There are relatively few reports of chlamydial infections in raptors in the United States [11–14]. In 1983, *C. psittaci* was isolated from four red-tailed hawks (*Buteo jamaicensis*; RTHA) in northern California (CA) and in 1992, *C. psittaci* was cultured from a RTHA in Louisiana with respiratory distress and diarrhea [11,12]. A recent study reported a 1.37% prevalence of chlamydial DNA in free-ranging raptors in northern CA, although anti-chlamydial antibodies using an elementary body agglutination (EBA) assay were not identified in any bird [14]. In Oregon, 3.6% of raptors at rehabilitation centers were positive for *C. psittaci* DNA [13]. In 2019 a red-shouldered hawk (*Buteo lineatus*; RSHA), one of 12 birds in an enclosure, developed conjunctivitis and died. Antemortem and postmortem testing including whole genome sequencing identified a novel chlamydial organism with *C. psittaci* and *C. abortus* as closest relatives and it was proposed that this novel organism be identified as *C. buteonis*, RSHA strain [10].

*Chlamydiaceae* species have significant impacts on human and animal health worldwide [15,16]. Recent discoveries of new species such as *C. buteonis* highlights major gaps in our understanding of host range, diversity and pathogenicity of *Chlamydiaceae* family members. Continued characterization of novel chlamydial organisms provides evidence that additional species may be pathogenic and pose zoonotic risks [15,17]. Recent expansions in their known host range highlight the risk for spill-over infections with increasing contact between wildlife, livestock, and humans because of greater wildlife habitat encroachment, intensification of livestock production, and pet ownership [15,17]. Most human cases of *C. psittaci* have been linked to companion birds, but some are linked to free-living birds [18,19]. A *C. psittaci* genotype identified from a raptor at a Belgian refuge was also cultured from three human contacts; one person reported symptoms consistent with chlamydiosis [19]. *Chlamydia psittaci* can be transmitted via ingestion or inhalation of ocular and nasal discharges and droppings from infected birds leading to risk of outbreaks at rehabilitation centers [20–22]. However, for many chlamydial species, modes of transmission are still unclear.

Diagnosis of chlamydial infections is challenging due to the absence or variability of clinical signs; multi-modal diagnostic testing is recommended for confirmatory diagnosis [23]. Isolation in cell culture was considered the gold standard until recently; now PCR-based detection is considered the gold standard for chlamydial diagnostics [24,25]. Samples are collected from epithelial surfaces that shed *C. psittaci* (e.g., conjunctiva, choana, and cloaca) or from infected tissues such as liver, lung, air sac, and spleen [23,26,27]. Additional testing includes serology using paired serum samples collected from acute- and convalescent- phase patients obtained at least 2 weeks apart and tested in the same laboratory at the same time [23]. Commercially available EBA assays detect *C. psittaci* IgM, while indirect fluorescent antibody tests (IFA) detect *Chlamydia* spp. IgY at the genus level [23]. As a result, serology can lead to conflicting results based on chlamydial species and time of sample collection.

To our knowledge there are no studies assessing detection of chlamydial DNA and seroprevalence in raptors admitted to wildlife rehabilitation centers in the US. Additionally, investigations of infection risk factors are lacking in wild birds. The objectives of this study were to evaluate the prevalence of chlamydial DNA shedding and seroprevalence in raptors admitted

to five wildlife rehabilitation centers throughout CA over a one-year period and to explore risk factors associated with infection.

## Materials and methods

### Ethics statement

This study was approved by the Institute of Animal Care and Use Committee of the University of California, Davis (protocol #19119).

### Sample collection

All raptors presenting to five CA wildlife rehabilitation centers from April 2016 to May 2017 were sampled. The following centers participated in the study and represented northern, central and southern CA: (1) California Raptor Center (CRC; Davis, northern CA), (2) Pacific Wildlife Care (PWC; Morro Bay, central CA), (3) Monterey SPCA (MSPCA; Monterey, central CA), (4) Living Desert Zoo Rehabilitation Center (LD; Palm Desert, southern CA), and (5) California Wildlife Center (CWC; Calabasas, southern CA). Criteria for inclusion included permissions from California Department of Fish and Wildlife to collect diagnostic samples from raptors, an on-site veterinarian for sample collection and an average intake of 200+ raptors/year. Veterinarians performed a full physical examination on each raptor at admission and recorded their findings, admission date, location found, species, sex, age class, and reason for admission. Conjunctival, choanal, and cloacal (c/c/c) mucosal swabs were obtained using a sterile, rayon-tipped swab (Puritan, 25–806 1PR). The swab tip was placed in a sterile 1.5 ml Eppendorf tube and stored at -20˚ C until shipment. Every three months, swabs were shipped overnight to the Infectious Disease Laboratory, University of Georgia (IDL) for quantitative real-time polymerase chain reaction (qPCR) to detect *Chlamydia* spp. DNA. Whole blood was collected from the jugular or medial metatarsal veins. Blood samples were placed into lithium heparin tubes and centrifuged at 3800 x g for 6 minutes. Plasma was collected into two cryovials and stored at -20˚C pending shipment. Every three months, plasma was shipped overnight to the University of California, Davis and stored at -80˚C until overnight shipment to the IDL. One aliquot of plasma was tested for chlamydial antibodies using elementary bodies purified from a RTHA (IDL17_4553_RTH) and Swainson's hawk [*Buteo swainsonii*; SWHA]) (IDL16-5840_SH) hawk *Chlamydia buteonis* isolate at the IDL. A second aliquot was shipped to the Avian and Wildlife Laboratory, University of Miami (AWL) for *Chlamydia* genus-specific IFA.

Two birds (one RTHA [(IDL17_4553_RTH)] and one SWHA [IDL16-5840_SH]) admitted to the CRC exhibited clinical signs suggestive of chlamydial infection. Both birds were positive by *Chlamydia* spp. DNA via qPCR and subsequently euthanized. Necropsies were performed within 12 hours. Lung, liver, spleen and air-sac tissues were collected for culture using aseptic technique. Tissues were placed in cryovials with sterile saline, and shipped overnight on ice to the IDL for chlamydial culture [27,28].

### Culture

Tissue samples from two birds were disrupted on the Mini Bead Mill (VWR) using 1.5ml tubes pre-filled with 2.4mm metal beads (VWR 10158–598). Filtration using a 0.45 um filter syringe was performed on tissue homogenates and inoculated on Vero-cell monolayers with DMEM (Corning 10-013-CV) supplemented with non-essential amino acids and 10% fetal calf serum according to previously published protocols [3,27]. Cell cultures with suggestive cytopathic effects (CPE) were tested for chlamydial DNA via qPCR. Briefly, 1uL of heat-inactivated cell culture supernatant was tested for *Chlamydia* spp. DNA via the ompA qPCR described previously

[27,28]. Positive cultures from the SWHA (IDL16-5840_SH) and RTHA (IDL17_4553_RTH) went through six passages to obtain nearly 100% CPE to maximize the concentration of infectious elementary bodies (EB) for isolation. DNA was extracted from purified EBs using a nucleic acid purification extraction (QuickExtract™, Epicentre Biotechnologies, Madison, WI) according to manufacturer's instructions and frozen at -80°C until sequencing.

## Sequencing and bioinformatic analysis

Chlamydial DNA was submitted for high throughput sequencing to the Centers for Disease Control and Prevention. The NEB Next Ultra DNA library prep kit for Illumina (New England Biolabs, Ipswich, MA) was used to prepare gDNA libraries according to the manufacturer's protocol. Genome sequencing was performed on the isolates using the Illumina MiSeq desktop sequencer (Illumina, San Diego, CA). Illumina sequencing read quality was evaluated using FastQC program (v0.10.1) [29]. Processing was performed with Cutadapt (v1.5) [30] and reads were removed from the data set if they met one of the following criteria: (a) had low quality (< 25) sequence bases; (b) trimmed adapter sequences; (c) had an error rate above 0.03; or (d) less than 75 base pairs in length. De novo assembly was performed with VelvetOptimiser v2.2.5 and Velvet v1.2.10 [31]. Long read sequencing was performed using the Pacific Biosciences (PacBio) RS II (Pacific Biosciences, Menlo Park, CA). One or two SMRT cells were used for each isolate with a movie time of 240 minutes. The sequences were assembled with SMRT Analysis v2.2 Hierarchical Genome Assembly Process version 2 (HGAP 2) or 3 (HGAP 3) protocol. The Illumina sequencing data associated with each isolate was aligned to the long read PacBio assembly for nucleotide accuracy comparison. Average nucleotide identity (ANI) was determined using program Pyani (v0.2.10) with default settings [32]. A maximum-likelihood phylogenomic tree was computed using IQTree [33] + ModelFinder [34] with 1000 iterations of bootstrapping from a pan-genomic analysis using Anvi'o [35] based on aligned protein sequences from all shared single copy gene clusters. Anvi'o software identified 742 single copy genes that were distinguishable as homologous among all genomes and used for further phylogenetic analysis. Sequences (SWHA (IDL16-5840_SH) and RTHA (IDL17_4553_RTH) were submitted to NCBI BioProject accession number PRJNA613727.

## Quantitative PCR

Swabs were processed for DNA extraction by boiling samples for 5 minutes in buffered saline solution containing dithiothreitol (DTT). Initially, standard target DNA amplification of the *ompA* sequence for *C. psittaci* qPCR with melting curve analysis was executed using the LightCycler 480 System and probe detection as previously described [27,36]. Although positive samples were amplified using *C. psittaci* primers, the melting curve analysis of the amplicon showed that the curve Tm and shape was unique from *C. psittaci* positive-control samples, thus the samples were determined as "atypical". To further identify the "atypical" positive samples, extracted DNA was amplified by targeted PCR for *C. buteonis* using primers 17473F (`AGCTCACATCAT CGCTCTCG`) and 17937R (`TGCGTGTTGTCGAACTAGCT`) followed by gel electrophoresis. Further, Sanger sequencing was performed by Genewiz (South Plainfield, NJ) to confirm the identity of *C. buteonis* PCR products, which aligned to *C. buteonis* with 100% identity.

## Serology

All plasma samples were evaluated for antibodies using isolate-specific EBA serology [36]. The EBs from RTHA (IDL17-4553) were obtained by centrifuging infected Vero cells at 46,467 x G for 1 h. The pellet was washed in Opti-MEM (Thermo Fisher, Grand Island, NY) and centrifuged at 1,200 x G for 5 min. The EBs were inactivated, layered on renografin, and centrifuged at 60,226 x G for 90 min. The band containing EBs was attained, pelleted, and resuspended in

diluent to a working concentration as described previously [36]. A suspension of EBs was mixed with unknown plasma samples on a slide by a mechanical rotator for 3 minutes at 150 rpm [37]. Samples were seropositive if agglutination was present [37]. Antibody titers were recorded as the highest serum dilution with agglutination, if there was no agglutination in control wells. EBA titers ≥1:10 were considered positive.

The second aliquot of plasma was shipped to AWL for IFA IgY testing. Fixed slides with the *C. trachomatis* inclusion stage (previously grown in McCoy cells on IFA slides under 5% $CO_2$ and 37°C in Modified Eagle's medium (MEM) with supplements) were stored at -70°C and evaluated using a previously published method [38]. A titer >1:25 was considered positive [38].

### Statistical analysis

The prevalence of raptors with antibodies against *Chlamydia* spp. and chlamydial DNA detection by qPCR were estimated overall and for each center separately. Associations between *Chlamydia* spp. infection, serological status and putative risk factors for infection were analyzed using Chi-square tests of independence. Risk factors evaluated included: (1) species; (2) age class: nestling (nestling, pre-fledge, and branchling), hatch year/juvenile (fledglings and 1st year birds), and adult, (3) season of admission—fall (September–November), winter (December–February), spring (March–May), and summer (June–September), (4) rehabilitation center, and (5) region (northern, central, and southern CA). Demographic and environmental variables associated with infection or seropositivity (p ≤ 0.1) were further evaluated using multivariable logistic regression models. Sex was excluded from the multivariable analyses as it was undetermined for the majority (n = 196) of birds. Potentially confounding variables and interactions were evaluated in the models. A variable was considered to be confounding if there was a ≥ 10% change between the adjusted and unadjusted estimates of the odds ratios for other variables in the model. Final parsimonious models were selected by comparing Akaike's Information Criterion (AIC) among competing models. Adjusted odds ratios with 95% confidence intervals were estimated to assess the strength of the association between risk factors and chlamydial status. All statistical tests were performed with R statistical software version 3.4.2 [39].

### Temporal and spatial statistical analyses

Locations at which the birds were found were manually geocoded from zip code data using Google Earth version 5.2 (Google Inc, Mountain View, California, USA). Latitude and longitude coordinates recorded using the WGS 84 geographic coordinate system were converted to decimal degrees. Clustering of seropositive birds and birds testing positive for chlamydial DNA was evaluated using the Bernoulli model and purely spatial and purely temporal scan statistics in SatScan™ version 9.0 [40]. Locations of sampled birds and statistically significant clusters were mapped using ArcGIS® (ESRI, Redlands, California, USA).

## Results

Chlamydial DNA was sampled from 263 raptors representing 18 species. The number of birds sampled from each rehabilitation center was: CRC (n = 43), MSPCA (n = 54), PWC (n = 79), CWC (n = 53) and LD (n = 34).

### Sequencing and bioinformatic analysis

The sequencing results were nearly identical to the recently described *C. buteonis* sequence isolated from a RSHA [10]. Phylogenetic analysis revealed that the organisms cultured from SWHA (IDL16-5840) and RTHA (IDL17-4553) were most similar to the recently described *C.*

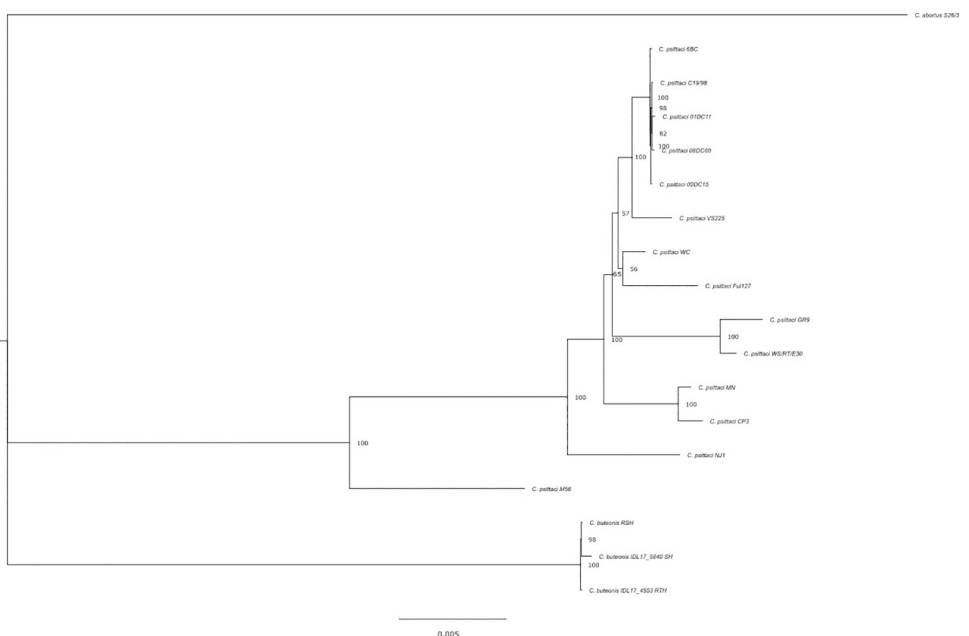

**Fig 1. Maximum likelihood phylogenetic tree showing that the SWHA and RTHA isolates were most similar to *C. buteonis*.** The tree was computed using programs IQtree [33]+ ModelFinder [34] with 1000 iterations of bootstrapping from a pan-genomic analysis using Anvi'o [35] based on aligned protein sequences from all shared sing copy gene clusters.

*buteonis* sequence. This analysis supported the inclusion of the SWHA and RTHA isolates in the clade containing *C. buteonis* as determined by Laroucau et al. [10], which is a phylogenetically intermediate between *C. psittaci* and *C. abortus* (Fig 1); comparison between the recently described *C. buteonis* sequence [10] and the SWHA and RTHA isolates revealed 99.97% ANI.

## Quantitative PCR

Swab specimens from 11/263 (4.18%) raptors were positive on *Chlamydia* spp. qPCR but were not characteristic for *C. psittaci* and were identified as "atypical positive;" all 11 positive chlamydial samples were also positive using *C. buteonis* primers (Table 1). Sanger sequencing from qPCR positive amplicons were identical to the recently described *C. buteonis* sequence [10]. Positive qPCR samples were identified in 6.4% of qPCR positive birds taxonomically classified in the family Accipitridae and 2.0% in the family Strigidae (Table 1). Further, the most detected qPCR positive species was RTHA (n = 8), followed by SWHA (n = 1), Cooper's hawk (*Accipiter cooperii*; COHA) (n = 1), and Great horned owl (*Bubo virginianus*; GHOW) (n = 1) (Table 1). Although positive qPCR results were more commonly detected in RTHAs, differences in prevalence across species and taxonomic families were not statistically significant (p = 0.798, p = 0.060, respectively). Additionally, positive qPCR prevalence was similar across age classes including 3.04% hatch years (n = 8), 0.38% juveniles (n = 1), and 0.76% adults (n = 2) (Table 2). However, significant seasonal differences for positive qPCR samples were identified (p = 0.002). The greatest number of positive birds were admitted in winter (n = 7), followed by spring (n = 2), summer (n = 2), and fall (n = 0) (Table 2). Birds admitted in the winter season were 8.1 times more likely to be qPCR positive than birds admitted during other seasons (OR = 8.1, 95% CI: 2.3–29.1).

Positive qPCR samples were identified from 3/43 (6.98%) birds admitted to CRC (2 RTHAs, 1 SWHA), 4/54 (7.41%) birds from MSPCA (4 RTHAs), 3/79 (3.80%) birds from

**Table 1. Results for chlamydial DNA and serology (elementary body agglutination; EBA and immunofluorescent antibody; IFA) by raptor species.**

| Family | Bird Species | qPCR | | EBA | | IFA | |
|---|---|---|---|---|---|---|---|
| | | n | Positive | n | Positive | n | Positive |
| Accipitridae | Bald Eagle | 1 | 0 | 1 | 0 | 1 | 0 |
| | Cooper's hawk | 43 | 1 (2.3%) | 34 | 1 (2.9%) | 34 | 0 |
| | Rough-legged hawk | 1 | 0 | 1 | 0 | 1 | 0 |
| | Red-shouldered hawk | 27 | 0 | 19 | 0 | 18 | 0 |
| | Red-tailed hawk | 70 | 8 (11.4%) | 56 | 10 (17.9%) | 55 | 9 (16.4%) |
| | Swainson's hawk | 11 | 1 (9.0%) | 5 | 1 (20.0%) | 6 | 0 |
| | White-tailed kite | 2 | 0 | 2 | 0 | 2 | 0 |
| | NA-hawk | 1 | 0 | 1 | 0 | 1 | 0 |
| | **Subtotal** | **156** | **10 (6.4%)** | **119** | **12 (10.1%)** | **118** | **9 (7.6%)** |
| Cathartidae | Turkey Vulture | 9 | 0 | 7 | 1 | 7 | 0 |
| | **Subtotal** | **9** | **0 (0.0%)** | **7** | **1 (14.3%)** | **7** | **0 (0.0%)** |
| Falconidae | American Kestrel | 4 | 0 | 2 | 0 | 2 | 0 |
| | Merlin | 1 | 0 | 0 | 0 | 0 | 0 |
| | Peregrine falcon | 3 | 0 | 2 | 0 | 1 | 0 |
| | Prairie falcon | 1 | 0 | 1 | 1 (100.0%) | 1 | 1 (100.0%) |
| | **Subtotal** | **9** | **0 (0.0%)** | **5** | **1 (20.0%)** | **4** | **1 (25.0%)** |
| Pandionidae | Osprey | 1 | 0 | 0 | 0 | 0 | 0 |
| | **Subtotal** | **1** | **0 (0.0%)** | **0** | **0 (0.0%)** | **0** | **0 (0.0%)** |
| Strigidae | Burrowing owl | 2 | 0 | 0 | 0 | 0 | 0 |
| | Great horned owl | 46 | 1 (2.2%) | 36 | 1 (2.7%) | 35 | 1 (2.8%) |
| | Long-eared owl | 1 | 0 | 0 | 0 | 0 | 0 |
| | Western screech owl | 1 | 0 | 1 | 0 | 1 | 0 |
| | **Subtotal** | **50** | **1 (2.0%)** | **37** | **1 (2.7%)** | **36** | **1 (2.8%)** |
| Tytonidae | Barn owl | 38 | 0 | 22 | 0 | 21 | 0 |
| | **Subtotal** | **38** | **0 (0.0%)** | **22** | **0 (0.0%)** | **21** | **0 (0.0%)** |

Numbers of positive birds for each species are presented with the percentage testing positive in parenthesis below the number.

PWC (1 Great Horned Owl [*Bubo virginianus*; GHOW], 2 RTHAs), and 1/34 (2.94%) birds from LD (1 Cooper's hawk (*Accipiter cooperii*; COHA; Table 2). No *C. buteonis* qPCR positives were detected in birds admitted to CWC. This corresponded to prevalence estimates of 6.98% (3/43) in northern CA, 5.26% (7/133) in central CA, and 1.15% (1/87) in southern CA; however, the differences across centers or regions were not statistically significant based on bivariate analyses (p = 0.318, p = 0.190, respectively).

A significant spatial cluster of raptors positive for *C. buteonis* DNA by qPCR (p = 0.002) was detected in northern CA, extending 24.5 km from its center at 38.310500 N, 121.901800 W and encompassing communities between Sacramento and San Francisco (Fig 2). A significant temporal cluster of raptors testing positive for *C. buteonis* DNA was detected during the winter season lending further support to the seasonal pattern of infections.

## Serology

In total, 15 of 190 (7.89%) plasma samples were positive for isolate-specific anti-chlamydial EBA antibodies. Two birds positive via EBA had *Chlamydia* spp. qPCR positive results (Table 1). Complementary to the qPCR results, the EBA and IFA seroprevalences did not differ by taxonomic family (p = 0.245, p = 0.229, respectively). In the family Accipitridae, 10.1% of

**Table 2. Comparative risk factors for raptors for positive qPCR (n = 263), elementary body agglutination antibodies (EBA; n = 190), or immunofluorescent antibodies (IFA; n = 186) with species, age class, wildlife rehabilitation center, and season of admission.**

| | Patient Information | | | | | Test Results | | |
|---|---|---|---|---|---|---|---|---|
| Bird ID | Center | Species | Season | Sex | Age | qPCR | EBA | IFA |
| 62 | MSPCA | RTHA | W | NA | HY | + | NA | NA |
| 63 | MSPCA | RTHA | W | NA | HY | + | NA | NA |
| 64 | MSPCA | RTHA | W | NA | A | + | − | + |
| 65 | MSPCA | RTHA | W | NA | A | + | + | + |
| 226 | PWC | GHOW | S | NA | HY | + | − | − |
| 232 | PWC | RTHA | S | M | HY | − | + | − |
| 236 | PWC | RTHA | S | NA | HY | − | + | − |
| 246 | PWC | RTHA | W | NA | HY | + | + | − |
| 254 | PWC | GHOW | F | NA | HY | − | + | − |
| 263 | PWC | TUVU | F | NA | HY | − | + | − |
| 303 | PWC | RTHA | P | M | HY | − | NA | + |
| 307 | PWC | GHOW | P | M | HY | − | − | + |
| 341 | PWC | RTHA | W | NA | HY | + | − | NA |
| 412 | LD | COHA | S | NA | HY | + | − | − |
| 644 | CWC | SWHA | F | NA | A | − | + | − |
| 804 | CRC | COHA | S | NA | NA | − | + | − |
| 811 | CRC | RTHA | S | F | A | − | + | + |
| 813 | CRC | RTHA | S | NA | HY | − | − | + |
| 815 | CRC | RTHA | S | NA | HY | − | + | + |
| 816 | CRC | RTHA | S | M | A | − | + | + |
| 824 | CRC | RTHA | P | NA | A | − | + | + |
| 826 | CRC | RTHA | W | NA | A | − | + | + |
| 861 | CRC | PRFA | F | F | A | − | + | − |
| 931 | CRC | RTHA | P | NA | A | − | + | − |
| 1001 | CRC | RTHA | P | M | HY | + | − | + |
| 1002 | CRC | SWHA | P | F | J | + | NA | NA |
| 1003 | CRC | RTHA | W | M | HY | + | NA | NA |

Sex was excluded as this variable was undetermined for the majority (n = 196) of birds.

MSPCA: Monterey SPCA, PWC: Pacific Wildlife Center, LD: Living Desert, CRC: California Raptor Center. RTHA: Red-tailed hawk; GHOW: Great horned owl;

TUVU: Turkey vulture; COHA: Cooper's hawk; SWHA: Swainson's hawk; PRFA: Prairie falcon. F = Fall, W = Winter, P = Spring, S = Summer. HY = Hatch year,

J = Juvenile, A = Adult. +: Positive result, -: Negative result, NA: Test was not performed/result was not determined.

EBA samples were positive and 14.3% in the family Cathartidae, 20.0% in the family Falconidae and 2.7% in the family Strigidae were also positive (Table 1). Meanwhile, IFA results showed 7.6% of EBA positive birds were classified in the family Accipitridae, 25% in the family Falconidae, and 2.8% in the family Strigidae (Table 1). The most seropositive species via EBA and IFA was the RTHA (n = 10 and 9, respectively), followed by SWHA (n = 1 and 0 respectively), COHA (n = 1 and 0 respectively), Turkey vulture (*Cathartes aura*; TUVU) (n = 1 and 0 respectively), Prairie falcon (*Falco mexicanus Schlegel*; PRFA) (n = 1 and 1 respectively), and GHOW (n = 1 and 1 respectively), (Table 1). Seroprevalence by EBA and IFA varied according to age (p = 0.006, p = 0.035, respectively) with the highest seroprevalence in adult birds (EBA: 50.0% adults, 42.8% hatch years/juveniles, 0% nestlings; IFA: 54.5% adults, 45.4% hatch years/ juveniles, 0% nestlings). Adult birds were 3.5 times more likely to be seropositive by EBA (OR = 3.5; 95% CI: 1.1, 11.4) than juvenile birds (hatch years and nestlings). Age class was not

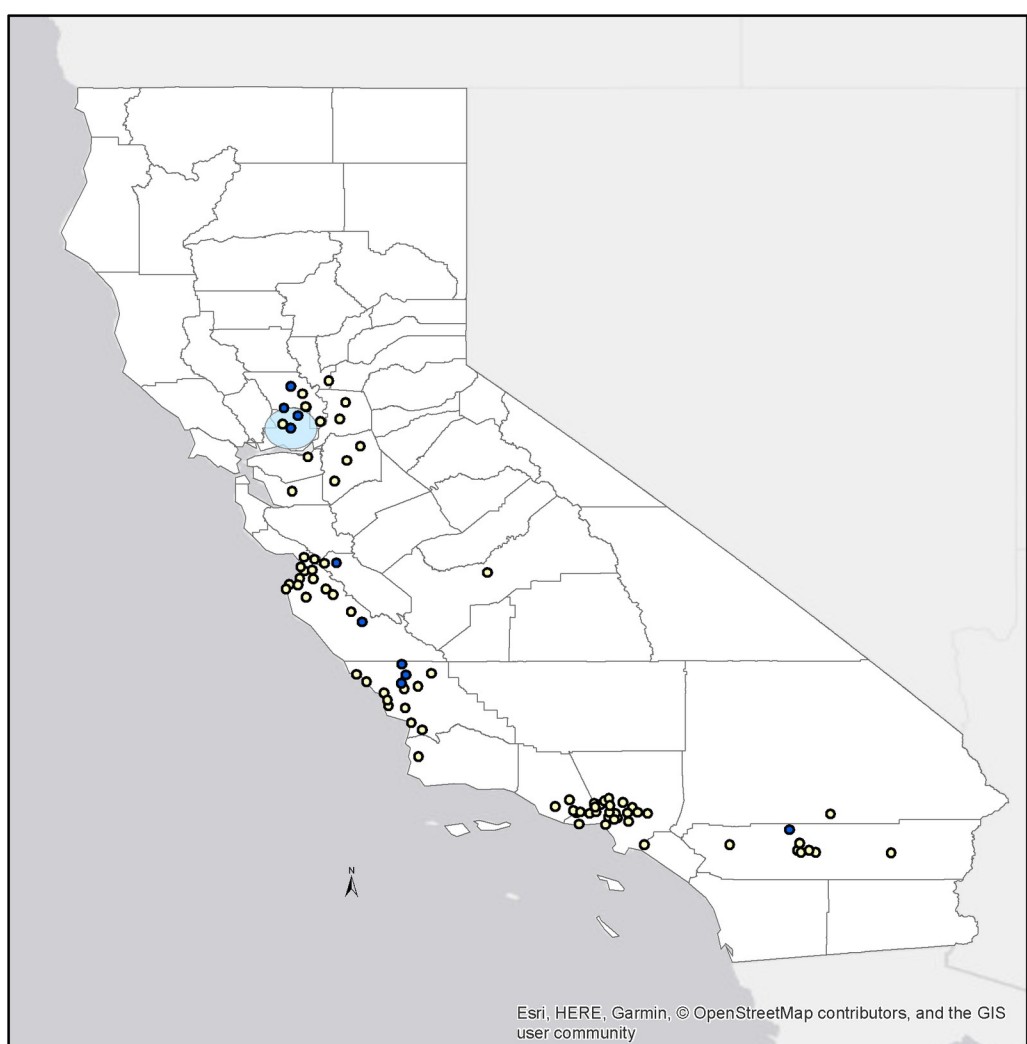

**Fig 2. Map of the significant spatial cluster for raptors testing positive for *Chlamydia* spp. DNA by rt-qPCR in northern California.** Distribution of raptors testing positive (blue) and negative (yellow; Center, 38.310500 N, 121.901800 W).

statistically significant in the logistic regression model for seropositivity by IFA; however, age class was included in the final IFA model to adjust for confounding. Seroprevalence by EBA and IFA were not variable by season (p = 0.776, p = 0.412, respectively).

Eight of 28 (28.57%) birds from the CRC, 1/14 (7.14%) birds from MSPCA, 5/74 (6.76%) birds from PWC, 0/53 (0%) birds from CWC, and 0/21 (0%) birds from LD were EBA sero-positive (Table 1). Differences in seroprevalence by EBA varied significantly across centers and regions (both p < 0.001) with the highest seropositivity in northern CA (28.57%), then central CA (6.82%), and southern CA (1.35%). Birds admitted to centers in northern CA were 5.6 times more likely to be seropositive by EBA (OR = 5.6; 95% CI: 1.7, 18.9) than other regions. Eleven of 186 (5.91%) birds were seropositive by *Chlamydia* genus-specific IFA (Table 2). Positive IFA results from 6/11 samples correlated with positive EBA results and three positive IFA birds had positive *Chlamydia* spp. qPCR positive results (Table 2). One qPCR positive RTHA was also positive for both EBA and IFA antibodies (Table 2).

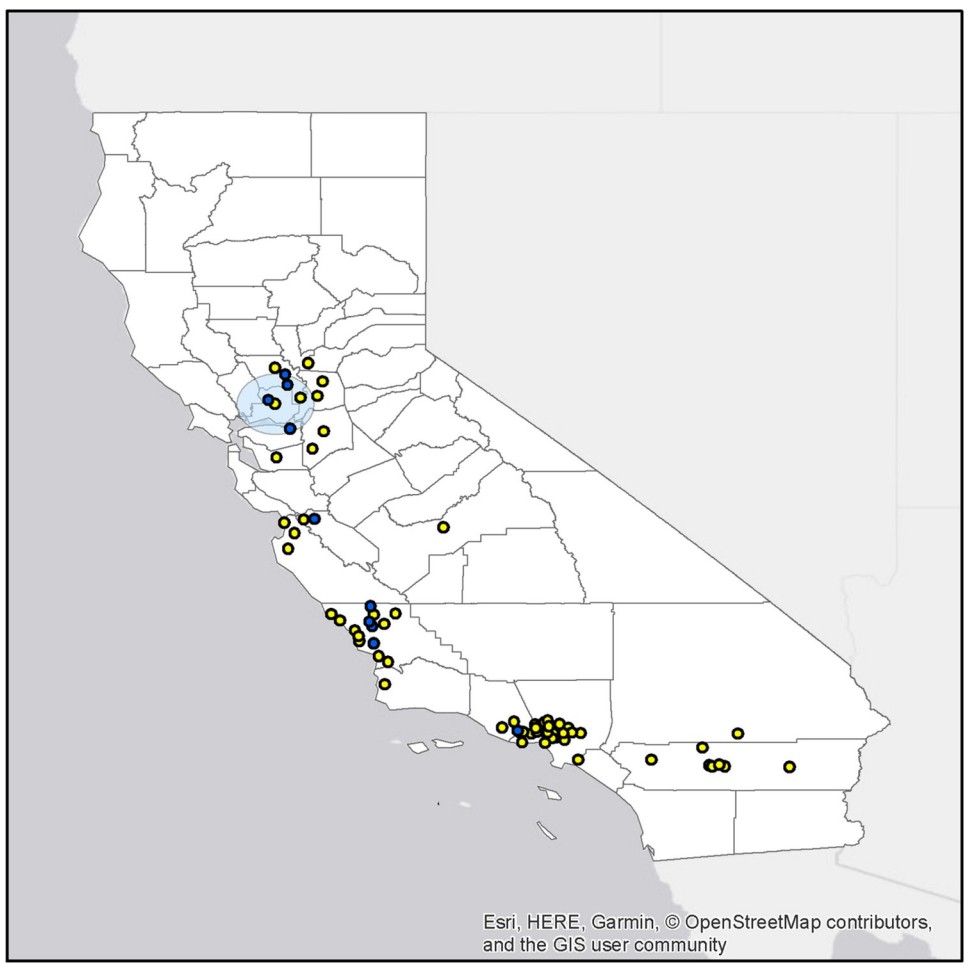

**Fig 3. Map of the significant spatial cluster of *Chlamydia* spp. seropositive raptors by EBA and/or IFA in northern California.** Distribution of seropositive raptors (blue) and seronegative raptors (yellow; Center, 38.310500 N, 121.901800 W).

Seroprevalence of IFA by rehabilitation center was: CRC: 7/28 (25.00%), MSPCA: 2/14 (14.29%), PWC: 2/72 (2.78%), CWC: 0/51 (0%), and LD: 0/21 (0%). IFA seroprevalences were estimated to be 25% in northern CA, 4.65% in central CA and 0% in southern CA. Differences in prevalence of anti-chlamydial antibodies by IFA across centers and regions were statistically significant (both p < 0.001). Birds in northern CA were 11.3 times more likely to be seropositive by IFA (OR = 11.3; 95% CI: 2.8, 45.3) than birds in other regions.

A significant spatial cluster of seropositive raptors (EBA and/or IFA) was detected (p = 0.001) in the same area (center at 38.310500 N, 121.901800 W) as the qPCR positive raptors, but encompassed a larger area, extending 42.40 km from its center (Fig 3). A significant temporal cluster of raptors testing positive for chlamydial DNA was detected during the winter season lending further support to the seasonal pattern of infections.

Nine of 11 (81.82%) birds positive by qPCR had physical examination findings recorded upon presentation. Clinical manifestations in infected birds were non-specific and included dehydration (8/9 birds) and co-infection with feather lice (5/9 birds). Most EBA seropositive birds were dehydrated (9/14) and all seropositive birds had a variety of traumatic injuries (e.g., wing injuries) and co-infections including coccidiosis, trichomoniasis, and avian pox.

## Discussion

Chlamydial DNA prevalence in raptors admitted to rehabilitation centers across CA was identified as "atypical" using *C. psittaci* primers and targeted PCR along with Sanger sequencing confirmed *C. buteonis* from the qPCR "atypical" positive DNA samples. Further, whole genome sequencing analysis performed on EBs from bacterial isolates from two raptors confirmed that the chlamydial isolate had a 99.97% ANI to *C. buteonis*, a newly identified species [10]. This is notable as the RSHA isolate was identified in France, whereas these isolates were identified in CA, USA. Isolate-specific EBA methodology was performed to investigate for IgM serological prevalence. One COHA and one GHOW had positive qPCR and antibody results; neither bird is in the genus *Buteo*. While previous research reported that C. buteonis has only been identified in the Buteo genus (RSHA), this study demonstrates that *C. buteonis* DNA was detected in other raptor hosts including a GHOW and detectable antibodies were identified in a TUVU, PRFA, and GHOW. These findings highlight the need for additional studies to evaluate diversity as well as pathogenicity and zoonotic potential of chlamydial species infecting raptors.

Chlamydial qPCR prevalence reported here (4.18%) is similar to data from raptors admitted to Oregon wildlife rehabilitation centers [13], and is higher than that found in free-ranging *Buteo* spp. in CA [14]. Similarities in prevalence between these studies could be explained by similar geographical locations of study populations or prey species available in Oregon and northern/central CA. Chlamydial qPCR prevalence was similar between free-ranging raptors in Sweden (1.3%) and northern CA (1.37%) [14,41]. The majority of raptors admitted to rehabilitation centers are sick or injured, thus chlamydial spp. infections would be expected to be higher than in free-ranging, presumably healthy birds. Meanwhile, a lower seroprevalence was detected in the study reported here than from raptors in Germany (63%) and previously from northern CA (44%) [11,42]; however, different testing methodologies for *C. psittaci* were utilized in those studies. The findings reported here may also differ from previous studies due to time of infection, immune response or handling artifacts. Moreover, a final sample size of 250 birds/center was anticipated based on previous intake numbers. However, centers reported lower intake numbers during the study leading to a smaller sample size of approximately 25% of expected.

Most birds qPCR-positive for *Chlamydia* DNA had negative serological results. This could result from detection of exogenous DNA from environmental contamination, plasma samples acquired before an immune response could be mounted or immunosuppression due to co-infections, toxins, drought or lack of prey. Because a single swab from multiple mucosal surfaces was analyzed for chlamydial DNA, the primary source of the organism cannot be determined. Multiple birds that were EBA seronegative were IFA seropositive, possibly due to birds being in the chronic stage of infection [28].

There were significant spatial clusters of *Chlamydia* DNA positive and seropositive birds detected in northern CA. It is possible that this chlamydial organism is confined to populations of raptors in northern, and to a lesser degree, central CA. An alternative explanation is that raptors are exposed to chlamydial organisms from infected dietary items that are more commonly found in these regions. In addition, chlamydial DNA prevalence and seroprevalence was highest in, but not exclusive to, RTHAs. Red-tailed hawks are one of the most abundant raptors within the CA central valley and constituted 26.6% of the sample population. The primary RTHA diet in the CA central valley consists of ground squirrels (*Spermophilus* spp.) black-tailed jackrabbits (*Lepus californicus*) and pocket gophers (*Geomys* spp., *Thomomys* spp.), although other mammals, small birds and reptiles have been reported [43]. It is possible that free-ranging rodents or rabbits may be a source of a chlamydial organism capable of

infecting raptors. To the authors´ knowledge, there are no reports confirming chlamydial organisms in these prey species.

Chlamydial qPCR positive prevalence was highest in birds presented during the winter, and seroprevalence increased with age. The higher prevalence in winter months could be associated with environmental stressors leading to reduced immunity. For example, rehabilitators often receive juvenile raptors emaciated with secondary infections during the winter [44]. It is also possible that raptors are increasingly exposed to the organism due to prey shifts in winter months. Birds testing positive for chlamydial DNA presented with nonspecific clinical signs, and all seropositive birds suffered from traumatic injuries and/or co-infections. Oregon rehabilitation centers reported similar co-morbidities with positive chlamydial DNA birds [13]. Positive birds reported here had common conditions such as pediculosis and dehydration; however, these conditions are common in ill raptors, regardless of the underlying condition. Correlations of clinicopathologic test results with *Chlamydia*-specific testing would be beneficial. Further epidemiological studies are needed to better understand hosts and transmission of this organism.

To our knowledge, this is the first prospective study assessing chlamydial DNA prevalence and seroprevalence in raptors admitted to wildlife rehabilitation centers in the US. Understanding the prevalence of this organism in rehabilitation centers is important because of its potentially high transmission rate to other birds and its relationship to other *Chlamydia* spp. with known zoonotic potential. Chlamydial infections in raptors can be problematic for individuals in contact with birds, including rehabilitation center staff, wildlife biologists, veterinarians and falconers [20,21]. Suspected zoonotic transmission of a chlamydial infection to volunteers at a rehabilitation center highlighted that wildlife rehabilitation workers were unaware of the symptoms of chlamydiosis [19]. Understanding the prevalence and risk factors associated with this pathogen aids in education and outreach to at-risk individuals regarding its zoonotic potential and prevention. The detection of atypical chlamydial organisms in wild raptors could also be of importance in the management of threatened or endangered species. Chlamydial bacteria can spread to other avian species, including sympatric avian populations such as SWHAs, a threatened raptor population in CA.

## Conclusion

This study reports the prevalence and risk factors associated with a recently characterized chlamydial species in raptors presenting to wildlife rehabilitation centers in CA, USA. Previous research identified *C. buteonis* in a RSHA; however, this study shows that *C. buteonis* has a larger host range that expands to other raptor families in addition to Accipitridae including Falconidae, Strigidae and Cathartidae. Information generated through this study will enhance awareness among wildlife rehabilitation staff of chlamydial infections in raptors and guide in assessments of admitted birds. These results demonstrate that the current understanding of chlamydial infections in raptors is limited and additional studies are needed to fully elucidate the diversity and epidemiology of *Chlamydia* spp. circulating among raptor populations in CA.

## Acknowledgments

The authors thank the veterinarians and staff from the following rehabilitation centers for their sample collection and processing: California Raptor Center, California Wildlife Center, Pacific Wildlife Care, Monterey SPCA, Living Desert.

## Author Contributions

**Conceptualization:** Brittany A. Seibert, Branson W. Ritchie, Michelle G. Hawkins.

**Data curation:** Brittany A. Seibert, Michael K. Keel, Roger A. Nilsen, Michelle G. Hawkins.

**Formal analysis:** Brittany A. Seibert, Terra R. Kelly, Roger A. Nilsen, Paula Ciembor, Denise Pesti, Christopher R. Gregory, Branson W. Ritchie, Michelle G. Hawkins.

**Funding acquisition:** Brittany A. Seibert, Michelle G. Hawkins.

**Investigation:** Brittany A. Seibert, Christopher R. Gregory, Branson W. Ritchie, Michelle G. Hawkins.

**Methodology:** Brittany A. Seibert, Michael K. Keel, Terra R. Kelly, Roger A. Nilsen, Paula Ciembor, Denise Pesti, Christopher R. Gregory, Branson W. Ritchie, Michelle G. Hawkins.

**Project administration:** Branson W. Ritchie, Michelle G. Hawkins.

**Resources:** Michelle G. Hawkins.

**Supervision:** Michelle G. Hawkins.

**Validation:** Roger A. Nilsen, Paula Ciembor, Denise Pesti, Christopher R. Gregory.

**Writing – original draft:** Brittany A. Seibert, Michael K. Keel, Terra R. Kelly, Roger A. Nilsen, Branson W. Ritchie, Michelle G. Hawkins.

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
