## [Decision Letter · Decision Letter 0]

7 Sep 2021

PONE-D-21-20941Chlamydia buteonis in birds of prey presented to California wildlife rehabilitation facilitiesPLOS ONE

Dear Dr. Hawkins,

Thank you for submitting your manuscript to PLOS ONE. After careful consideration, we feel that it has merit but does not fully meet PLOS ONE’s publication criteria as it currently stands. Therefore, we invite you to submit a revised version of the manuscript that addresses the points raised during the review process. Please address each issue raised by the reviewers in a point-by-point response with corrections to the text/tables/figures where indicated.

Please submit your revised manuscript within Oct 22 2021 11:59PM.  If you will need more time than this to complete your revisions, please reply to this message or contact the journal office at plosone@plos.org. Please include the following items when submitting your revised manuscript:A rebuttal letter that responds to each point raised by the academic editor and reviewer(s). You should upload this letter as a separate file labeled 'Response to Reviewers'.A marked-up copy of your manuscript that highlights changes made to the original version. You should upload this as a separate file labeled 'Revised Manuscript with Track Changes'.An unmarked version of your revised paper without tracked changes. You should upload this as a separate file labeled 'Manuscript'.If applicable, we recommend that you deposit your laboratory protocols in protocols.io to enhance the reproducibility of your results. Protocols.io assigns your protocol its own identifier (DOI) so that it can be cited independently in the future. For instructions see: https://journals.plos.org/plosone/s/submission-guidelines#loc-laboratory-protocols. Additionally, PLOS ONE offers an option for publishing peer-reviewed Lab Protocol articles, which describe protocols hosted on protocols.io. Read more information on sharing protocols at https://plos.org/protocols?utm_medium=editorial-email&utm_source=authorletters&utm_campaign=protocols.

We look forward to receiving your revised manuscript.

Kind regards,

Deborah Dean, M.D., M.P.H.

Academic Editor

PLOS ONE

2. Please provide methods of sacrifice in the Methods section of your manuscript

 “This study was financially supported through the Karen C. Drayer Wildlife Health Center Fellowship Grant Program, School of Veterinary Medicine, Univ. of California, Davis, DAvis, CA.”

“The authors thank the Wildlife Health Center Fellowship grant program for their financial support of the study and the veterinarians and staff from the following rehabilitation centers for their sample collection and processing: California Raptor Center, California Wildlife Center, Pacific Wildlife Care, Monterey SPCA, Living Desert. We would also like to thank the Infectious Disease Laboratory, University of GA for the generous donation of the qPCR testing, EBA testing and chlamydial cultures and the Avian and Wildlife Laboratory, University of Miami for partial donation of IFA testing.”

“This study was financially supported through the Karen C. Drayer Wildlife Health Center Fellowship Grant Program, School of Veterinary Medicine, Univ. of California, Davis, DAvis, CA.”

7. We noted in your submission details that a portion of your manuscript may have been presented or published elsewhere. [DETAILS AS NEEDED] Please clarify whether this [conference proceeding or publication] was peer-reviewed and formally published. If this work was previously peer-reviewed and published, in the cover letter please provide the reason that this work does not constitute dual publication and should be included in the current manuscript.

8. Please upload a new copy of Figure 1 as the detail is not clear. Please follow the link for more information: " ext-link-type="uri" xlink:type="simple">https://blogs.plos.org/plos/2019/06/looking-good-tips-for-creating-your-plos-figures-graphics/"
https://blogs.plos.org/plos/2019/06/looking-good-tips-for-creating-your-plos-figures-graphics/.

Reviewers' comments:

Reviewer's Responses to Questions

**Comments to the Author**

1. Is the manuscript technically sound, and do the data support the conclusions?

Reviewer #1: Yes

Reviewer #2: Yes

2. Has the statistical analysis been performed appropriately and rigorously? 

Reviewer #1: Yes

Reviewer #2: Yes

3. Have the authors made all data underlying the findings in their manuscript fully available?

Reviewer #1: Yes

Reviewer #2: Yes

4. Is the manuscript presented in an intelligible fashion and written in standard English?

Reviewer #1: Yes

Reviewer #2: Yes

5. Review Comments to the Author

Reviewer #1: The findings of this study are new and interesting as they appear shortly after publication of the species Chlamydia buteonis. Samples for serology and qPCR were collected from 263 birds of prey, representing 18 species. Chlamydial DNA was found in 4.18 % and anti-chlamydial Abs detected in 3.14 % of tested birds. As could be expected, a serologic response was typically observed in adults. Nevertheless, the inclusion of serology in parallel to DNA testing is an important asset of the present work. The manuscript is concise and well written. Data processing and presentation are adequate.

A few points remain unclear and should be revised.

1. Line 114: Plasma samples were divided into two aliquots and tested using two serologic assays. Please, indicate the chlamydia species of the hawk isolate used in the assay at the IDL.

2. Line151-52:

a) It is unclear from the text whether both positive cultures from SWHA (IDL16-

5840_SH) and RTHA (IDL17_4553_RTH) were genome sequenced or only one. How many sequences will be (or have been) deposited at NCBI?

b) It seems that complete assembly of the raw sequences has not been achieved as 742 clusters are reported. If so, a reason should be given.

3. Lines 157-59: The qPCR assay for C. buteonis used here is probably new and has not been published previously. If so, the authors should include data showing the specificity of that assay. Besides, the protocol of a specific and sensitive assay can be found in reference 10.

4. Line 207: The authors claim that the clade containing their own C. buteonis isolates is distinctly different from other chlamydial species. However, there is only one more species, C. psittaci, on Fig. 1. The substantiate the statement, at least one more species should appear in the figure, e.g. C. trachomatis as a non-avian Chlamydia sp.

5. In reference 10, only hawks (genus Buteo) were mentioned as known hosts of C. buteonis. The present study revealed new observations on host specificity of this species that should be summarized and included in the text as a major conclusion.

Reviewer #2: The authors present an analysis of detection of chlamydial DNA and/or antibodies in raptors presenting to wildlife rehabilitation centres. The analysis is clear and easy to follow, and the trends apparent related to age, different species are also clearly documented. The sample size of 263 from 18 species is good size, and across several distinct centres. The evidence of increasing serological responses in age and chlamydial DNA associated with disease supports the conclusion of the potential for risk and the prevalence of the organism in these wild populations. The sequence analysis confirming similarity to a recently described c.buteonis adds confidence to the findings. Overall this study is a valid and useful addition to the literature on chlamydia in wildlife.

6. PLOS authors have the option to publish the peer review history of their article (what does this mean?). If published, this will include your full peer review and any attached files.

Reviewer #1: No

Reviewer #2: **Yes: **Willa Huston

---

## [Author Response · Author response to Decision Letter 0]

20 Sep 2021

Responses to Editorial Comments:

1. We have worked to ensure that all of the formatting conforms to PlosOne. 

2. No animals were “sacrificed,” therefore nothing was included in the methods regarding this. 

3. I apologize for not initially providing complete information on the minimum dataset, as this is my first PlosOne submission. I have reviewed the entire manuscript and all data necessary to complete statistical analyses are included within the manuscript in the tables and figures. We have no ethical or legal issues with sharing our data. Specifically these 5 items are addressed:

1) Sequences (SWHA (IDL16-5840_SH) and RTHA (IDL17_4553_RTH) were submitted to NCBI BioProject accession number PRJNA613727 (Lines 174, 175 in manuscript)

2) Data for the primers for the qPCR for C. buteonis are within the manuscript (Line 186, 187 in manuscript)

3) Fig 1. Maximum likelihood phylogenetic tree showing that the SWHA and RTHA isolates were most similar to C. buteonis. 

4) Tables 1 and 2 provide the raw data for chlamydial DNA and serology (elementary body agglutination; EBA and immunofluorescent antibody; IFA) by raptor species, therefore providing the raw data used for the statistical analyses. 

5) Fig 2 and 3. Map of the significant spatial cluster for raptors testing positive for Chlamydia spp. DNA by rt-qPCR in northern California was developed using methods described within the manuscript and using the raw data using the qPCR results from table 2.

4. It is stated that the grant information provided in the ‘Funding Information’ and ‘Financial Disclosure’ sections do not match. I updated the “funding information” section but it was not a grant, rather in-kind donation of the qPCR, EBA serology and chlamydial cultures. As several authors are from this laboratory I was unsure how else to clarify? Our new financial disclosure should read “This study was financially supported through the Karen C. Drayer Wildlife Health Center Fellowship Grant Program, School of Veterinary Medicine, Univ. of California, Davis, Davis, CA and the Infectious Disease Laboratory, College of Veterinary Medicine, University of Georgia, Athens, GA.”

5. I would like to update the information regarding the role of funders for this research. The Karen C. Drayer Wildlife Health Center funders had no role in study design, data collection and analysis, decision to publish, or preparation of the manuscript. The Infectious Disease Laboratory, University of GA donated the costs of the qPCR testing, EBA testing and chlamydial cultures and were instrumental in data analysis, decision to publish and preparation of the manuscript. Multiple authors are affiliated with the laboratory (Ritchie, Nilsen, Ciembor, Pesti, Gregory) but only Ritchie and Nilsen were involved in manuscript preparation. We have removed the Avian and Wildlife Laboratory at the Univ. of Miami, as this is a research price given to all researchers and therefore we do not believe now is a source of funding. Please let me know if you disagree?

6. Acknowledgements of funding within the manuscript have been removed (Lines 447-449). The corrected statement reads: “The authors thank the veterinarians and staff from the following rehabilitation centers for their sample collection and processing: California Raptor Center, California Wildlife Center, Pacific Wildlife Care, Monterey SPCA, Living Desert. 

7. This work was presented at the 2016 ExoticsCon conference in Portland, OR by my Master’s student Seibert (first author) and was published for association members in their proceedings for that year as a 250 word abstract. The abstract is only reviewed by one person for approval to present at the conference. This is the abstract “Chlamydia psittaci is a reportable, zoonotic bacterium with health implications for birds and humans. Due to its zoonotic potential, veterinary staff should be updated on the prevalence and clinical presentations of Chlamydia spp. infections in raptors. A retrospective study evaluating raptors testing positive for Chlamydial infections was conducted through the UC Davis Veterinary Medical Teaching Hospital medical records from 2004 –2016. A confirmed or probable case was based on the National Association of Public Health Veterinarians criteria for C. psittaci.1 Specific C. psittaci diagnostic testing included real-time PCR (RT-PCR) from a combination mucosal swab, elementary body agglutination testing, indirect fluorescent antibody testing, immunofluorescent antibody staining for Chlamydia spp. antigen, and isolation of the organism. Signalment (sex, age, seasonality and species) and clinicopathologic results from positive cases were analyzed. A Chi-squared test was used to statistically evaluate the data, and significance was defined as p ≤ 0.05. Forty eight red-tailed hawks (RTHA; Buteo jamaicensis) and 1 Swainson’s hawk (B. swainsoni) were identified as confirmed or probable cases. A 22.9% prevalence was identified from the total number of raptors tested for C. psittaci and 1.9% prevalence was identified from the total number of all raptors admitted to the VMTH. There was a significant difference in prevalence between age (p0.005), seasonality (p0.001) and species (p0.001). Tissues most commonly identified as positive were lung, liver, air sac, heart and kidney. Genetic sequencing of tissue samples from two RTHA with conflicting IFA/EBA results suggests that these infections may be caused by a novel Chlamydial-like organism.” We did not feel this 250 word abstract for a conference constituted previous publication. 

8. I put figure 1 into the PACE system and it did not find any image problems but did increase the dpi to 300 x 300 and created a proper TIF. I hope this is acceptable? Please let me know if it is not.

9. We reviewed our references. None have been retracted. 

Responses to Reviewers:

Thank you for giving us the opportunity to re-submit a revised manuscript of “Chlamydia buteonis in birds of prey presented to California wildlife rehabilitation facilities” for PlosOne. We appreciate yours along with the reviewers’ time and effort spent providing feedback and insightful comments to help improve our manuscript. We have incorporated most of the suggestions made by you and the reviewers. Please see below a point-by-point response to the reviewers’ comments and concerns. Line numbers refer to the revised manuscript file with tracked changes. 

5. Review Comments to the Author

Reviewer #1: The findings of this study are new and interesting as they appear shortly after publication of the species Chlamydia buteonis. Samples for serology and qPCR were collected from 263 birds of prey, representing 18 species. Chlamydial DNA was found in 4.18 % and anti-chlamydial Abs detected in 3.14 % of tested birds. As could be expected, a serologic response was typically observed in adults. Nevertheless, the inclusion of serology in parallel to DNA testing is an important asset of the present work. The manuscript is concise and well written. Data processing and presentation are adequate.

A few points remain unclear and should be revised.

1. Line 114: Plasma samples were divided into two aliquots and tested using two serologic assays. Please, indicate the chlamydia species of the hawk isolate used in the assay at the IDL.

Author Response: Thank you, the species of the hawk isolates used were added. 

2. Line151-52:

a) It is unclear from the text whether both positive cultures from SWHA (IDL16-

5840_SH) and RTHA (IDL17_4553_RTH) were genome sequenced or only one. How many sequences will be (or have been) deposited at NCBI?

Author Response: Thank you, both positive cultures were sequenced, and this information was added. Two sequences have been deposited at NCBI under the BioProject accession number PRJNA613727. 

b) It seems that complete assembly of the raw sequences has not been achieved as 742 clusters are reported. If so, a reason should be given.

Author Response: Thank you, the Anvi'o software determined 742 single copy genes that were distinguishable as homologous between all the genomes and determined to be single copy genes in all the genomes; and therefore, used for the phylogenetic analysis. This was not a measure of completeness of any genome, just the number of genes that were used in the phylogenetic analysis. The description of the methods was updated in the manuscript. 

3. Lines 157-59: The qPCR assay for C. buteonis used here is probably new and has not been published previously. If so, the authors should include data showing the specificity of that assay. Besides, the protocol of a specific and sensitive assay can be found in reference 10.

Author Response: Thank you, we expanded on the flow of the C. buteonis identification in the positive samples within the method section. The samples were first analysed using the C. psittaci specific qPCR with a melting curve analysis following a previously described method. However, the positive samples from this study showed a different melting curve signature compared to C. psittaci positive control samples. Thus, those samples were amplified using C. buteonis specific primers described in the method section and then Sanger sequenced to confirm C. buteonis identity. 

4. Line 207: The authors claim that the clade containing their own C. buteonis isolates is distinctly different from other chlamydial species. However, there is only one more species, C. psittaci, on Fig. 1. To substantiate the statement, at least one more species should appear in the figure, e.g. C. trachomatis as a non-avian Chlamydia sp.

Author Response: Thank you, we understand that the conclusion suggested results showing more than 1 species and therefore edited the results section of Figure 1 to state the following “This analysis supported the inclusion of the SWHA and RTHA isolates in the clade containing C. buteonis as determined by Laroucau et al, which is a phylogenetically intermediate between C. psittaci and C. abortus.”

5. In reference 10, only hawks (genus Buteo) were mentioned as known hosts of C. buteonis. The present study revealed new observations on host specificity of this species that should be summarized and included in the text as a major conclusion.

Author Response: Thank you. This finding was emphasized in the abstract, discussion and conclusion. 

Reviewer #2: The authors present an analysis of detection of chlamydial DNA and/or antibodies in raptors presenting to wildlife rehabilitation centres. The analysis is clear and easy to follow, and the trends apparent related to age, different species are also clearly documented. The sample size of 263 from 18 species is good size, and across several distinct centres. The evidence of increasing serological responses in age and chlamydial DNA associated with disease supports the conclusion of the potential for risk and the prevalence of the organism in these wild populations. The sequence analysis confirming similarity to a recently described c.buteonis adds confidence to the findings. Overall this study is a valid and useful addition to the literature on chlamydia in wildlife.

Thank you so much for your review.

---

## [Editor Report · Decision Letter 1]

29 Sep 2021

Chlamydia buteonis in birds of prey presented to California wildlife rehabilitation facilities

PONE-D-21-20941R1

Dear Dr. Hawkins,

We’re pleased to inform you that your manuscript has been judged scientifically suitable for publication and will be formally accepted for publication once it meets all outstanding technical requirements.

Kind regards,

Deborah Dean, M.D., M.P.H.

Academic Editor

PLOS ONE
---

## [Editor Report · Acceptance letter]

4 Oct 2021

PONE-D-21-20941R1 

*Chlamydia buteonis* in birds of prey presented to California wildlife rehabilitation facilities 

Dear Dr. Hawkins:

I'm pleased to inform you that your manuscript has been deemed suitable for publication in PLOS ONE. Congratulations! Your manuscript is now with our production department. 

Kind regards, 

on behalf of

Dr. Deborah Dean 

Academic Editor

PLOS ONE